# Anisotropic RANS Turbulence Modeling for Wakes in an Active Ocean Environment [†]

**Dylan Wall** *[,‡,§] **and Eric Paterson** [‡,§]

Department of Aerospace and Ocean Engineering, Virginia Polytechnic Institute and State University, Blacksburg, VA 24061, USA; egp@vt.edu

* Correspondence: dylanjw@vt.edu; Tel.: +1-865-816-0632
† This paper is an extended version of our paper published in 15th OpenFOAM Workshop.
‡ Current address: Randolph Hall, RM 215, 460 Old Turner Street, Blacksburg, VA 24061, USA.
§ These authors contributed equally to this work.

**Abstract:** The problem of simulating wakes in a stratified oceanic environment with active background turbulence is considered. Anisotropic RANS turbulence models are tested against laboratory and eddy-resolving models of the problem. An important aspect of our work is to acknowledge that the environment is not quiescent; therefore, additional sources are included in the models to provide a non-zero background turbulence. The RANS models are found to reproduce some key features from the eddy-resolving and laboratory descriptions of the problem. Tests using the freestream sources show the intuitive result that background turbulence causes more rapid wake growth and decay.

**Keywords:** stratified wakes; turbulence; RANS; stress-transport

---

## 1. Introduction

Oceanographic flows include a broad variety of turbulence-generating phenomena, and the associated unsteady motions are in general inhomogeneous, non-stationary, and anisotropic. The thermohaline stratification of the ocean introduces a conservative body force which must be considered when examining flows in such an environment. Numerous effects including buoyancy, shear, near-free-surface damping, bubbles, and Langmuir circulations complicate any attempt to describe turbulent motions. The variety of production mechanisms includes (but is not limited to) wind shear, wave breaking, internal gravity waves, double diffusion, and overturning due to the alternating heating and cooling of the ocean surface.

The numerical simulation of engineering-related problems in such an environment is a daunting prospect. For the case of wakes generated by ships and other man-made objects, the associated Reynolds number can be $\mathcal{O}(10^9)$ in the near field, while approaching very small Froude number in the far field. The broad separation of scales means that the use of scale-resolving methods such as large-eddy simulation (LES) or direct numerical simulation (DNS) at full scale is typically prohibitively computationally expensive for use in design and analysis problems. In previous work, we have tested a set of Reynolds stress models (RSMs) against laboratory representations of the oceanographic environment (see Wall and Paterson [1], publication pending). In this work, we then further develop the application of these models to the problem of wakes, testing against laboratory and scale-resolving model descriptions of stratified wakes. The models are then modified with source terms to produce a finite background turbulence, intended to model the environmental turbulence in the ocean.

As has been noted, one of the key complications associated with the ocean is the density stratification, which causes anisotropy in the stress tensor. In the late wake, the effects of the

stratification inevitably dominate the character of turbulent motion. Models which account for buoyancy-induced stress tensor anisotropy in some way are therefore desirable for simulations of late-wake behavior, necessitating a second-moment closure. There is an extensive history of second-moment approaches in modeling geophysical problems (early examples are the hierarchy of Mellor and Yamada [2] or the summary of Rodi [3]). Similar approaches have also been adopted in dealing with stratified wakes. Algebraic closures paired with two-equation models have been used to good effect by, for example, Hassid [4], Voropaeva et al. [5], or Voropaeva et al. [6]. Such models reproduce key behaviors of stratified wakes, including the vertical collapse and production of internal waves. Full stress-transport models have also been applied to the problem, with the earliest example being Lewellen et al. [7]. More recently, Voropaeva [8] has even adopted algebraic and transport-equation based triple-moment closures.

Due to the expectation of strong stratification effects in the late wake, models which remain realizable in an approximately two-component turbulence state are desirable. The family of cubic tensorial expansion models developed at the University of Manchester were developed with the two-component limit (TCL) as an explicit constraint, and have been applied to a variety of flows with strong buoyant forcing (for example, see Craft et al. [9], Craft and Launder [10], Suga [11], or Craft et al. [12]) The so called TCL model has also been applied to doubly-stratified (simultaneous salinity and temperature stratifications) environs, as presented by Armitage [13].

In evaluating RANS closures, it is possible to employ a wealth of LES and DNS numerical experiments which have been conducted to complement previous experimental studies of stratified wakes. Temporal-model, or 3D+t simulations such as those conducted by Dommermuth et al. [14] or Brucker and Sarkar (Brucker [15], Brucker and Sarkar [16]), have helped to describe the distribution of energy within the wakes of both towed and self-propelled bodies. More recently, body-inclusive simulations such as those conducted by Chongsiripinyo and Sarkar [17] have done much to refine the understanding the scaling laws which can be applied a given stratified wake, and to qualify each stage encountered during its life. These stages were originally identified as the three-dimensional (3D), non-equilibrium (NEQ), quasi-two-dimensional (Q2D), and viscous three-dimensional (V3D) stages by Spedding [18]. It is an interesting digression to note that these stages roughly align with the stages of a full-scale ship wake as defined by Somero et al. [19] (the near wake, the far wake, and the persistent wake), though the rigorous definitions are different in each case. Some recent experiments and LES/DNS studies have also concerned themselves with the effect of non-trivial free-stream turbulence. Studies such as that of Amoura et al. [20] and Rind and Castro [21] have shown that environmental turbulent motions can have dramatic effects on wake behavior. In the case of stratified turbulence, the simulations of Radko and Lewis [22] include consideration of pre-existing double-diffusive fluctuations. The authors of that study also establish fairly simple wake-detection criteria based on the centerline deficits of dissipation rate $\epsilon$ and turbulent scalar variance ($\overline{\theta^2}$ or $\overline{s^2}$). It is fully understood that much of the physics described by these scale-resolving simulations will be lost in adopting a RANS approach; the results of these studies must then be carefully applied to refining RANS models.

Having addressed the scope of the problem, we now recapitulate a set of general criteria which must be satisfied by a turbulence model which might be applied to full-scale, far-field ship wakes in an oceanic environment:

- The model must be implementable as part of a general-use computational fluid dynamics package (in the case of this work, the finite-volume code OpenFOAM)
- The model must be able to accommodate the anisotropy that arises in stratified turbulent flows. Paramount is accounting for anisotropy in the energy-containing eddies, however, under many stratification conditions anisotropy may also arise throughout the turbulence wavenumber spectrum (see, for example, Khani and Waite [23]).

- The model must gracefully handle free-stream environmental turbulence, with differing levels of turbulent variance in both the temperature and salinity fields. Ideally, the impact of pre-existing turbulence on wake similarity would be accurately accounted for as well.
- The model must reproduce classic stratified wake experiments such those of Lin and Pao [24]
- The model must reproduce key behaviors observed using scale resolving methods, including decay rates of turbulence kinetic energy (TKE) and turbulence potential energy (TPE), and the growth (or lack thereof) of the wake in the vertical and horizontal directions.

The novelty in the work presented here is primarily in the application and testing of stress transport RANS models to the wake problem, and the addition of sustaining turbulence sources to begin addressing the issue of environmental turbulence. As such, the simulations and evaluation described for this study were conducted primarily to address the third and fifth bullet points. The simulation approach (including computational methods and initial/boundary conditions) and turbulence model closure methods employed are detailed in Section 2. The results of comparisons between the RANS model predictions and laboratory/LES models of stratified wakes are provided in Section 3, which also includes some commentary on these results. Section 4 provides some brief concluding remarks and discusses avenues for further work.

## 2. Simulation Methodology

The model system of equations was solved using extensions to the open-source finite-volume fluid dynamics package OpenFOAM. A "2D + t" approach was adopted, the same as that employed by, for example, Lewellen et al. [25] or Voropaeva [8]. Mean field transport and RANS model equations were solved on a two-dimensional grid, representative of a slice of fluid through which the wake progenitor has passed.

### 2.1. Mean Transport Equations

Momentum was transported according to the Reynolds-averaged incompressible Navier–Stokes equations under the Boussinesq approximation:

$$\frac{\partial U_i}{\partial t} + U_j \frac{\partial U_i}{\partial x_j} = -\frac{1}{\rho_0} \frac{\partial P}{\partial x_i} + \frac{\rho - \rho_0}{\rho_0} g_i + \frac{\partial}{\partial x_j} \left( \nu \frac{\partial U_i}{\partial x_j} - \overline{u_i u_j} \right) \tag{1}$$

$$\frac{\partial U_i}{\partial x_i} = 0 \tag{2}$$

where $U_i$ is the mean-velocity vector, $u_i$ is the fluctuating component of velocity, $P$ is the mean kinematic pressure, $g_i$ is the gravitational vector, $\nu$ is the fluid viscosity, $\rho$ is the fluid density, and $\overline{u_i u_j}$ is the Reynolds stress tensor. For the laboratory-scale simulations conducted in this work, a linear equation of state for the density $\rho$ was deemed sufficient:

$$\frac{\rho - \rho_0}{\rho_0} = -\beta_S (S - S_0) - \beta_\Theta (\Theta - \Theta_0) \tag{3}$$

where the relevant scalar values are the mean temperature $\Theta$ and the mean salinity $S$, the 0 subscript denotes a reference value, and the expansion coefficients are defined by:

$$\beta_\Theta = -\frac{1}{\rho} \left. \frac{\partial \rho}{\partial \Theta} \right|_P, \qquad \beta_S = -\frac{1}{\rho} \left. \frac{\partial \rho}{\partial S} \right|_P \tag{4}$$

Note that all of the simulations presented in Section 3 are singly-stratified. In keeping with the methods employed for experimental study of stratified wakes, a salinity stratification was employed. The equation of state reduces to:

$$\frac{\rho - \rho_0}{\rho_0} = -\beta_S \left( S - S_0 \right) \tag{5}$$

For the remainder of the work, only the model for the transport of salinity will be provided. However, the same model can be applied as a temperature field under certain conditions. The scalar quantities are transported according to an advection–diffusion equation:

$$\frac{\partial \left( \delta S \right)}{\partial t} + U_i \frac{\partial \left( \delta S \right)}{\partial x_i} + U_i \frac{\partial S_B}{\partial x_i} = \frac{\partial}{\partial x_i} \left( \alpha_S \frac{\partial \left( \delta S \right)}{\partial x_i} - \overline{su_i} \right) \tag{6}$$

where the total mean scalar field $S$ is assumed to be the sum of a background $S_B$ and a perturbation to that background $\delta S$, and $\overline{su_i}$ is the turbulent flux of the scalar quantity. The quantities $\overline{u_i u_j}$ and $\overline{su_i}$ are supplied by the turbulence model.

## 2.2. Stress/Flux Transport

In general for this work, the framework and nomenclature for second moment models laid out by Hanjalić and Launder [26] is adopted, where mean quantities are denoted by capital symbols ($U$, $S$, etc.), fluctuating quantities by lower case symbols ($u$, $s$), and averaging is denoted by an over-bar ($\overline{u_i u_j}$, $\overline{su_i}$, $\overline{\theta^2}$). The stress tensor can be obtained by solving the associated transport equation:

$$\frac{\partial \overline{u_i u_j}}{\partial t} + U_k \frac{\partial \overline{u_i u_j}}{\partial x_k} = \mathcal{P}_{ij} + \mathcal{G}_{ij} + \Phi_{ij} - \epsilon_{ij} + \mathcal{D}_{ij} + \mathcal{P}_{ij_\infty} \tag{7}$$

where the terms on the right-hand side are the dissipation tensor $\epsilon_{ij}$, the shear production:

$$\mathcal{P}_{ij} = - \left( \overline{u_i u_k} \frac{\partial U_j}{\partial x_k} + \overline{u_j u_k} \frac{\partial U_i}{\partial x_k} \right) \tag{8}$$

the production due to the buoyancy body force in a Boussinesq fluid:

$$\mathcal{G}_{ij} = - \left( \mathcal{F}_j g_i + \mathcal{F}_i g_j \right) \tag{9}$$

$$\mathcal{F}_i = \beta_S \overline{su_i} + \beta_\Theta \overline{\theta u_i} \tag{10}$$

the re-distributive effects due to pressure interactions:

$$\Phi_{ij} = \overline{\frac{p}{\rho} \left( \frac{\partial u_i}{\partial x_j} + \frac{\partial u_j}{\partial x_i} \right)} \tag{11}$$

and diffusive effects:

$$\mathcal{D}_{ij} = \frac{\partial}{\partial x_k} \left[ \nu \frac{\partial \overline{u_i u_j}}{\partial x_k} - \overline{u_i u_j u_k} - \overline{\frac{p}{\rho} \left( u_i \delta_{jk} + u_j \delta_{ik} \right)} \right] \tag{12}$$

A free-stream sustaining source $\mathcal{P}_{ij_\infty}$ is also included, intended to maintain the TKE at some finite value (preferably associated with some background condition representative of the active ocean environment). The forms of the free-stream sources employed are given in Section 2.8. Similarly,

the transport of the turbulent flux of a scalar quantity, such as the temperature or salinity in ocean water, was described according to the equation:

$$\frac{\partial \overline{su_i}}{\partial t} + U_k \frac{\partial \overline{su_i}}{\partial x_k} = \mathcal{P}_{si}^S + \mathcal{P}_{si}^U + \mathcal{G}_{si} + \Phi_{si} + \mathcal{D}_{si} \tag{13}$$

where the physical interpretation of each term is the same as given for (7). The source terms are:

$$\mathcal{P}_{si}^S = -\overline{u_i u_j} \frac{\partial S}{\partial x_j} \tag{14}$$

$$\mathcal{P}_{si}^U = -\overline{su_j} \frac{\partial U_i}{\partial x_j} \tag{15}$$

$$\mathcal{G}_{si} = -\beta_S g_i \overline{s^2} \tag{16}$$

The terms $\epsilon_{ij}$, $\Phi_{ij}$, $\mathcal{D}_{ij}$, $\Phi_{sj}$, $\mathcal{D}_{sj}$, and the quantity $\overline{s^2}$ must be modeled in order to close the second-moment RSM. The following sections describe the closure approaches employed; the different models so constructed are summarized in Table 1.

**Table 1.** Summary of the model variations employed, with equation numbers. The models and implementation are described in detail in Section 2.

| Model | $\overline{u_i u_j}$ | $\epsilon$ | $\phi_{ij}$ | $\phi_{\theta i}$ | $\epsilon_{ij}$ | $\mathcal{P}_{ij_\infty}$ |
|-------|------|---|------|------|------|------|
| EVM1  | (45), (48) | (46), (47) | N/a | N/a | N/a | N/a |
| RSM1  | (7) | (39), (40) | (33)–(35) | (36)–(38) | (41) | (51) |
| RSM1a | (7) | (39), (40) | (33)–(35) | (36)–(38) | (42) | (51) |
| RSM1b | (7) | (39), (40) | (33)–(35) | (36)–(38) | (41) | (54) |
| RSM2  | (7) | (39), (40) | (27)–(29) | (30)–(32) | (41) | (51) |

### 2.3. Diffusive Process Closure

For all of the RSMs used in this work, the generalized gradient–diffusion hypothesis (GGDH) model originally proposed by Daly and Harlow [27] was employed to approximate the diffusive effects:

$$\mathcal{D}_{ij} = \frac{\partial}{\partial x_k} \left( \nu \frac{\partial \overline{u_i u_j}}{\partial x_k} - c_s \frac{k}{\epsilon} \overline{u_k u_l} \frac{\partial \overline{u_i u_j}}{\partial x_l} \right) \tag{17}$$

$$\mathcal{D}_{si} = \frac{\partial}{\partial x_k} \left( \gamma \frac{\partial \overline{\theta u_i}}{\partial x_k} - c_s \frac{k}{\epsilon} \overline{u_k u_l} \frac{\partial \overline{su_i}}{\partial x_l} \right) \tag{18}$$

Other, more complex closures for these terms have been applied stratified problems. The most pertinent example is the set of models employed by Voropaeva et al. [6], using both complex empirical algebraic expressions and even transport equations for a subset of the triple correlations. Craft and Launder [10] also recommend using transport equations which account for buoyancy effects on a subset of the triple correlations for strongly stratified flows. However, the effects described in that work were primarily associated with sharp pycnoclines, in which inhomogeneity in the triple correlations became significant. Such approaches would then likely be unnecessary for the linear-stratification environment in this work. Under certain environmental conditions, the diffusion closure may need further revision.

### 2.4. Pressure Strain/Scrambling Closure

In modeling the pressure strain and scrambling terms, it is common to adopt the approach of Chou [28], in which the pressure fluctuations are eliminated from (11) by taking the divergence of the transport equation for $u_i$ and so obtaining a Poisson equation for $p$. The details of such a derivation are here elided. The resulting expressions for $\Phi_{ij}$ and $\Phi_{si}$ can be arranged into terms associated

with different physical processes: the return to isotropy ($\Phi_1$), isotropization of mean strain ($\Phi_2$), isotropization of body forcing ($\Phi_3$), and wall blocking effects ($\Phi_w$). The wall blocking effects are neglected for this work, as the flow of interest is a free shear flow. The term (11) and analogous term from (13) can then be written as:

$$\Phi_{ij} = \Phi_{ij_1} + \Phi_{ij_2} + \Phi_{ij_3} \tag{19}$$

$$\Phi_{si} = \Phi_{si_1} + \Phi_{si_2} + \Phi_{si_3} \tag{20}$$

Typically, models make use of the stress anisotropy tensor $a_{ij}$ and its invariants; the associated definitions are included for completeness:

$$a_{ij} = \frac{\overline{u_i u_j}}{k} - \frac{2}{3}\delta_{ij} \tag{21}$$

$$A_2 = a_{ij}a_{ji} \tag{22}$$

$$A_3 = a_{ij}a_{jk}a_{ki} \tag{23}$$

Lumley's flatness parameter is also employed by some models:

$$A = 1 - \frac{9}{8}\left(A_2 - A_3\right) \tag{24}$$

which takes the value of unity in isotropic turbulence, and the value of zero in two-component turbulence. Additionally, the symmetric ($S$) and an antisymmetric ($W$) portions of the velocity gradient tensor are employed by, for example, the cubic pressure–strain model employed by RSM1 and the Boussinesq eddy–viscosity model:

$$S_{ij} = \frac{1}{2}\left(\frac{\partial U_i}{\partial x_j} + \frac{\partial U_j}{\partial x_i}\right) \tag{25}$$

$$W_{ij} = \frac{1}{2}\left(\frac{\partial U_i}{\partial x_j} - \frac{\partial U_j}{\partial x_i}\right) \tag{26}$$

Two pressure–strain models were employed for the simulations presented in this work. The simpler was a linear model, and the other a cubic model based on the work of Craft et al. [9]. The linear model employed the return-to-isotropy model first proposed by Rotta [29], and the linear isotropization-of-production terms from Launder et al. [30]:

$$\Phi_{ij_1} = -c_1 \epsilon a_{ij}, \qquad c_1 = 1.8 \tag{27}$$

$$\Phi_{ij_2} = -c_2\left(\mathcal{P}_{ij} - \frac{1}{3}\mathcal{P}_{kk}\delta_{ij}\right), \qquad c_2 = 0.6 \tag{28}$$

$$\Phi_{ij_3} = -c_3\left(\mathcal{G}_{ij} - \frac{1}{3}\mathcal{G}_{kk}\delta_{ij}\right), \qquad c_3 = 0.6 \tag{29}$$

The accompanying model of the pressure-scrambling processes in the scalar flux equations is detailed by Gibson and Launder [31]:

$$\Phi_{si_1} = -c_{1s}\frac{\epsilon}{k}\overline{su_i}, \qquad c_{1s} = 3.5 \tag{30}$$

$$\Phi_{si_1} = -c_{2s}\mathcal{P}_{si}^U, \qquad c_{2s} = 0.5 \tag{31}$$

$$\Phi_{si_1} = -c_{3s}\mathcal{G}_{si}, \qquad c_{3s} = 0.5 \tag{32}$$

The cubic model was that developed at the University of Manchester, and is detailed in the book by Hanjalić and Launder [26]. The model is designed to be realizable approaching the so-called two-component limit (TCL), at which one of the normal stresses is approximately zero. The pressure–strain process models were originally described by Craft et al. [9], and have here been re-cast in terms of the symmetric and antisymmetric portions of the velocity gradient tensor:

$$\Phi_{ij_1} = -c_1\left[a_{ij} + c_1'\left(a_{ik}a_{kj} - \frac{1}{3}A_2\delta_{ij}\right)\right] - \epsilon a_{ij}, \qquad c_1 = 3.1\,(A_2A)^{1/2},\ c_1' = 1.2 \tag{33}$$

$$
\begin{aligned}
\Phi_{ij_2} =& c_2 k\left(a_{ik}S_{jk} + a_{jk}S_{ik} - \frac{2}{3}a_{kl}S_{kl}\delta_{ij}\right) + c_3 k\left(a_{ik}W_{jk} - a_{jk}W_{ik}\right) + c_4 kS_{ij} + c_5 ka_{ij}\mathcal{P}_{kk} \\
&+ c_6 k\left(a_{ik}a_{kl}S_{jl} + a_{jk}a_{kl}S_{il} - 2a_{kj}a_{li}S_{kl} - 3a_{ij}a_{kl}S_{kl}\right) + c_7 k\left(a_{ik}a_{kl}W_{jl} + a_{jk}a_{kl}W_{il}\right) \\
&+ c_8 k\left[a_{mn}^2\left(a_{ik}W_{jk} + a_{jk}W_{ik}\right) + \frac{3}{2}a_{mi}a_{nj}\left(a_{mk}W_{nk} + a_{nk}W_{mk}\right)\right],
\end{aligned}
$$
$$c_2 = 0.6,\ c_3 = 0.866,\ c_4 = 0.8,\ c_5 = 0.3,\ c_6 = 0.2,\ c_7 = 0.2,\ c_8 = 1.2 \tag{34}$$

$$
\begin{aligned}
\Phi_{ij_3} =& -\left(\frac{3}{10} + \frac{3}{80}A_2\right)\left(\mathcal{G}_{ij} - \frac{1}{3}\delta_{ij}\mathcal{G}_{kk}\right) + \frac{1}{6}a_{ij}\mathcal{G}_{kk} \\
&+ \frac{2}{15}\mathcal{F}_m\left[g_i a_{mj} + g_j a_{mi}\right] - \frac{1}{3}g_k\left[a_{ik}\mathcal{F}_j + a_{jk}\mathcal{F}_i\right] \\
&+ \frac{1}{10}\delta_{ij}g_k a_{mk}\mathcal{F}_m + \frac{1}{4}g_k a_{mk}\mathcal{F}_m a_{ij} \\
&+ \frac{1}{8}g_k\left\{\mathcal{F}_m\left(a_{ki}a_{mj} + a_{kj}a_{mi}\right) - a_{mk}\left(a_{mj}\mathcal{F}_i + a_{mi}\mathcal{F}_j\right)\right\} \\
&- \frac{3}{40}\left\{a_{mk}\mathcal{F}_k\left(g_i a_{mj} + g_j a_{mi}\right) - \frac{2}{3}\delta_{ij}g_k a_{mk}a_{mn}\mathcal{F}_n\right\}
\end{aligned}
\tag{35}
$$

The pressure-scrambling process models were further developed by Craft and Launder [10], and for a doubly-stratified system (such as the temperature/salinity stratification in the ocean) by Armitage [13]:

$$
\begin{aligned}
\Phi_{si_1} =& -1.7\left(1 + 1.2\sqrt{A_2A}\right)r^{1/2}\frac{\epsilon}{k}\left(\overline{su_i}(1 + 0.6A_2) - 0.8a_{ik}\overline{su_k} + 1.1a_{ik}a_{kj}\overline{su_j}\right) \\
&- 0.2A^{1/2}rka_{ij}\frac{\partial S}{\partial x_j}
\end{aligned}
\tag{36}
$$

$$
\begin{aligned}
\Phi_{si_2} =& 0.8\overline{su_i}\frac{\partial U_i}{\partial x_k} - 0.2\frac{\partial U_k}{\partial x_i}\overline{su_k} + \frac{1}{6}\frac{\epsilon}{k}\overline{su_i}\frac{\mathcal{P}_{kk}}{\epsilon} \\
&- 0.4\overline{su_k}a_{il}\left(\frac{\partial U_m}{\partial x_l} + \frac{\partial U_l}{\partial x_m}\right) + 0.1\overline{su_k}a_{ik}a_{ml}\left(\frac{\partial U_k}{\partial x_l} + \frac{\partial U_l}{\partial x_k}\right) \\
&- 0.1\overline{su_k}\frac{1}{k}\left(a_{im}\mathcal{P}_{mk} + 2a_{mk}\mathcal{P}_{im}\right) + 0.15a_{ml}\left(\frac{\partial U_k}{\partial x_l} + \frac{\partial U_l}{\partial x_k}\right)\left(a_{mk}\overline{su_i} - a_{mi}\overline{su_k}\right) \\
&- 0.05a_{ml}\left[7a_{mk}\left(\overline{su_i}\frac{\partial U_k}{\partial x_l} + \overline{su_k}\frac{\partial U_i}{\partial x_l}\right) - \overline{su_k}\left(a_{ml}\frac{\partial U_i}{\partial x_k} + a_{mk}\frac{\partial U_i}{\partial x_l}\right)\right]
\end{aligned}
\tag{37}
$$

$$\Phi_{si_3} = -\frac{1}{3}\mathcal{G}_{si} - \beta_S \overline{s^2} a_{ik} g_k \tag{38}$$

Notably, the coefficients in (35), (37), and (38) are not empirical, and are determined only by the realizability constraints. The principle justification for adopting such a complex model is that, even for a wake with an initially high $Re$ and $Fr$, the flow will eventually decay to the point at which the turbulent Froude number $Fr_T = \epsilon/kN$ is small. The turbulence will then be dominated by stratification effects, and so approach the two-component limit. Recent LES/DNS studies such as those by Chongsiripinyo and Sarkar ([17,32]) have lent some credence to this notion, showing that the vertical normal stress decreases much more quickly than the horizontal normal stresses.

### 2.5. Scale-Equation Closure

The scale-determining equation is typically the most empirical part of a given RANS turbulence model. In describing stratified flows, a number of different quantities have been used to describe the scale of turbulent motion; the various transport equations for $kL$, $\epsilon$, or $\omega$ each have virtues, but are ultimately somewhat interchangeable (see Umlauf and Burchard [33]). For this work, the empirical model of the $\epsilon$ equation developed by Craft et al. [9] was adopted:

$$\frac{\partial \epsilon}{\partial t} + U_i \frac{\partial \epsilon}{\partial x_i} = \frac{\epsilon}{k}\left(\frac{1}{2}c_{\epsilon_1}\mathcal{P}_{kk} - c_{\epsilon_2}\epsilon + \frac{1}{2}c_{\epsilon_3}\mathcal{G}_{kk}\right) + \frac{\partial}{\partial x_i}\left[\left(\nu\delta_{ij} + c_\epsilon \frac{k}{\epsilon}\overline{u_i u_j}\right)\frac{\partial \epsilon}{\partial x_j}\right] + \mathcal{P}_{\epsilon_\infty} \tag{39}$$

As with the stress transport equation, a free-stream source $\mathcal{P}_{\epsilon_\infty}$ is included. For the stress transport models, the coefficients for the model $\epsilon$ transport Equation (39) were taken to be:

$$c_{\epsilon_1} = 1.0, \quad c_{\epsilon_2} = \frac{1.92}{1.0 + 0.7A_2^{1/2}A}, \quad c_{\epsilon_3} = 1.0, \quad c_\epsilon = 0.15 \tag{40}$$

The coefficient values and parameterizations in (40) are taken as-is from the works of, for example Craft and Launder [10], the combination of coefficients has been tested against a variety of free-shear flows (see Hanjalić and Launder [26] for a thorough accounting). The model for $\epsilon$ given by (39) and (40) is designed for free-shear flows, and has been found to be ill-posed for homogeneous turbulence (see Speziale [34]. It is therefore likely to be less-suited to the far-wake than to near-wake regions. Furthermore, Pereira and Rocha [35] has noted a general deficiency in models like (39) in the case of strongly-stratified turbulence. The empirical model $\epsilon$ equation is therefore perhaps the best target for model improvement in future work. Two different models for the dissipation rate tensor were tested. The first assumes that $\epsilon_{ij}$ is isotropic:

$$\epsilon_{ij} = \frac{2}{3}\delta_{ij}\epsilon \tag{41}$$

The second is the model of Hallbäck et al. [36], and adopts a nonlinear dependence on the anisotropy of the stress tensor:

$$\epsilon_{ij} = \epsilon\left[\frac{2}{3}\delta_{ij}(1 - f_s) + \frac{\overline{u_i u_j}}{k}f_s\right] - \frac{3}{4}\epsilon\left(a_{ik}a_{jk} - \frac{1}{3}A_2\delta_{ij}\right), \quad f_s = 1 + \frac{3}{4}\left(\frac{1}{2}A_2 - \frac{2}{3}\right) \tag{42}$$

### 2.6. Scalar-Variance Closure

Closure of (13) also requires the variance of the scalar fluctuations, $\overline{s^2}$. Per Radko and Lewis [22], $\overline{s^2}$ is also a potentially useful quantity in its own right. A transport equation can be solved for $\overline{s^2}$:

$$\frac{\partial \overline{s^2}}{\partial t} + U_i \frac{\partial \overline{s^2}}{\partial x_i} = -2\overline{su_i}\frac{\partial U_k}{\partial x_i} - \epsilon_{ss} + \frac{\partial}{\partial x_i}\left[\left(\alpha_S \delta_{ij} + c_{ss_s}\frac{k}{\epsilon}\overline{u_i u_j}\right)\frac{\partial \overline{s^2}}{\partial x_j}\right] + \mathcal{P}_{ss_\infty} \tag{43}$$

which includes a free-stream source $\mathcal{P}_{ss_\infty}$. Dissipation of $\overline{s^2}$ was modeled using the algebraic expression of Craft et al. [9]:

$$\epsilon_{ss} = r\frac{\epsilon}{k}\overline{s^2}, \qquad r = 1.5\left(1 + \frac{\overline{su_i}\,\overline{su_i}}{k\overline{s^2}}\right) \tag{44}$$

Some preliminary simulations conducted using a transport equation for $\epsilon_{ss}$, rather than (44), indicated that the algebraic expression was sufficient. However, if environmental conditions are expected to have significantly different scalar and mechanical time scales (fossilized turbulence, for example), this evaluation may need revision.

## 2.7. Eddy Viscosity Model

For the sake of comparison, the same set of wake conditions was also applied to an isotropic eddy–viscosity model. A standard $k$-$\epsilon$ model was employed, with the body-force effects on the scale equation modeled after the approach of Henkes et al. [37]:

$$\frac{\partial k}{\partial t} + U_i\frac{\partial k}{\partial x_i} = \mathcal{P}_k - \epsilon + \mathcal{G}_k + \frac{\partial}{\partial x_i}\left[\left(\nu + \frac{1}{\sigma_k}\nu_T\right)\frac{\partial k}{\partial x_j}\right] + \mathcal{P}_{k_\infty} \tag{45}$$

$$\frac{\partial \epsilon}{\partial t} + U_i\frac{\partial \epsilon}{\partial x_i} = \frac{\epsilon}{k}\left(c_{\epsilon_1}\mathcal{P}_k - c_{\epsilon_2}\epsilon + c_{\epsilon_3}\mathcal{G}_k\right) + \frac{\partial}{\partial x_i}\left[\left(\nu + \frac{1}{\sigma_\epsilon}\nu_T\right)\frac{\partial \epsilon}{\partial x_j}\right] + \mathcal{P}_{\epsilon_\infty} \tag{46}$$

$$c_{\epsilon_1} = 1.44, \quad c_{\epsilon_2} = 1.92, \quad c_{\epsilon_3} = \tanh\left(\frac{|U_3|}{\sqrt{U_1^2 + U_2^2}}\right)c_{\epsilon 1}, \quad \sigma_k = 1.4, \quad \sigma_\epsilon = 1.3 \tag{47}$$

As with (40), the coefficient values and parameterizations in (47) are taken and tested as given in the literature (Henkes et al. [37], in this case). The remaining modeled values are closed with the Boussinesq approximation:

$$\overline{u_iu_j} = \tfrac{2}{3}k\delta_{ij} - 2\nu_T S_{ij}, \qquad \overline{su_i} = -\nu_T\frac{\partial S}{\partial x_i} \tag{48}$$

$$\mathcal{P}_k = \overline{u_iu_j}\frac{\partial U_i}{\partial x_j}, \qquad \mathcal{G}_k = \beta_S g_i\overline{su_i} \tag{49}$$

$$\nu_T = C_\mu\frac{k^2}{\epsilon}, \quad C_\mu = 0.09 \tag{50}$$

## 2.8. Environmental Turbulence Sources

Finally, in order to accommodate the existence of environmentally generated background turbulence, free-stream source terms were introduced to the turbulence quantity transport equations, as proposed by Spalart and Rumsey [38]. The terms maintain the background turbulence quantities at a specified value, and have the added benefit of improving numerical stability and convergence. For the $\overline{u_iu_j}$, the simplest approach is to introduce an isotropic source (the $\infty$ subscript denotes a free-stream value):

$$\mathcal{P}_{ij_\infty} = \frac{2}{3}\delta_{ij}\epsilon_\infty \tag{51}$$

The sources for the $\epsilon$, $\overline{s^2}$ equations, respectively, are then:

$$\mathcal{P}_{\epsilon_\infty} = c_{\epsilon 2} \frac{\epsilon_\infty^2}{k_\infty} \tag{52}$$

$$\mathcal{P}_{ss_\infty} = 1.5 \frac{\epsilon_\infty}{k_\infty} \overline{s^2}_\infty \tag{53}$$

A second, nearly two-component anisotropic source, was also implemented, to test the potential impact of free-stream anisotropy:

$$\mathcal{P}_{ij_\infty} = \frac{2}{3} \begin{bmatrix} \frac{9}{10} & 0 & 0 \\ 0 & \frac{9}{10} & 0 \\ 0 & 0 & \frac{2}{10} \end{bmatrix} \epsilon_\infty \tag{54}$$

Note that, as the dissipation of the scalar flux $\overline{su_i}$ is typically included in the pressure-scrambling model, and any of the flux-vector components may potentially be negative, such source terms for the flux-transport equations were not applied. Unless otherwise stated, in the simulations conducted for this work, the free-stream values $k_\infty$ $\epsilon_\infty$ were chosen such that $\nu_{t_\infty} = C_\mu k_\infty^2 / \epsilon_\infty \approx 0.5\nu$ For the simulations presented in this work, these forms were employed. Further study may be needed to select forms which properly account for the stratification and background anisotropy.

*2.9. Simulation Approach*

As indicated in the introduction, the models were implemented for the open-source finite-volume code OpenFOAM. For all of the simulations detailed here, a second-order backward numerical differentiation scheme was applied for the temporal derivatives. Second-order linear schemes were applied for all spatial derivatives, with limiters applied as necessary to ensure convergence. The momentum Equation (1) and continuity Equation (2) were coupled using the widely employed hybrid PISO-SIMPLE (PIMPLE) algorithm, modified to include a Boussinesq body force after the fashion of Issa and Oliveira [39]. Note that, under the $2D + t$ approach employed, the axial component of the velocity $U_1$ given by (1) is actually the velocity difference:

$$U_1 = U_s = U_{1,total} - U_B \tag{55}$$

where $U_B$ is the propagation velocity of the wake generator. The mean velocity and turbulence kinetic energy were initialized according to idealized models of drag and self-propelled (or net-zero momentum, NZM) wakes. The wakes were initially axially symmetric, and the Reynolds stress tensor was initially isotropic. For the drag wake, the expressions for velocity and TKE, in terms of radial position $r$, were:

$$U_s = U_{s,CL,0} \exp\left[-\frac{1}{2}\left(\frac{r}{D}\right)^2\right] \tag{56}$$

$$k = k_{CL,0} \left[1 + 4\left(\frac{r}{D}\right)^2\right] \exp\left[-2\left(\frac{r}{D}\right)^2\right] \tag{57}$$

where the initial centerline values are dependent on the case in question. The form of (57) produces a TKE distribution roughly in line with the sphere wake measurements of Uberoi and Freymuth [40]. For the self-propelled (NZM) wake, the expressions for velocity and TKE were:

$$U_s = U_{s,CL,0} \left[1 - 2\left(\frac{r}{D}\right)^2\right] \exp\left[-2\left(\frac{r}{D}\right)^2\right] \tag{58}$$

$$k = k_{CL,0} \exp\left[-2\left(\frac{r}{D}\right)^2\right] \tag{59}$$

The form of (59) produces a TKE distribution which roughly resembles the measurements in the wake of the disk/jet wake of Naudascher [41]. The expressions for both wake types are plotted as a function of radial position in Figure 1. Note that the expression for the self-propelled wake produces a smaller total amount of TKE than that of the drag wake. For all cases, the dissipation rate $\epsilon$ was set such that the turbulent Reynolds number $Re_T = k^2/\nu\epsilon$ had a constant value of 10,000 throughout the wake. In studies employing scale-resolving methods (e.g., Dommermuth et al. [14] or Brucker and Sarkar [16]), it was found that initializing fluctuations in the scalar value did not substantially change the behavior of the wake. In applying the RSMs to the same problem, algebraic expressions were tested to initialize $\overline{su_i}$ and $\overline{s^2}$, based on assuming equilibrium for the transport equations of those quantities. However, use of the algebraic initialization was found to have little effect on the RSM predictions outside of the initial stage of wake development. As the LES studies simply initialized these quantities to zero, the scalar-associated turbulence quantities were therefore likewise initialized to zero for this study.

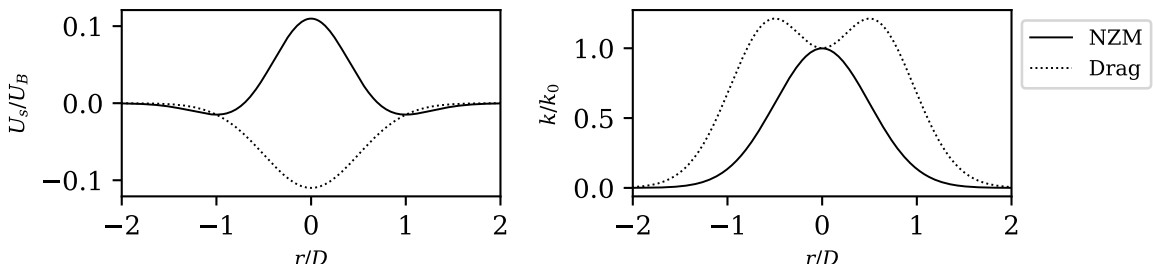

**Figure 1.** The initial radial distribution of axial velocity and TKE, given by Equations (56)–(59), for both a drag and a self-propelled wake.

The simulation domain consisted of a square two-dimensional grid, $120D$ in both vertical and horizontal extent. The pressure field $p$ was given a Dirichlet boundary condition, with a fixed value of zero. The other flow variables were given mixed Dirichlet/Neumann boundary conditions, dependent on the flux of the quantity at the boundary.

As is readily seen in Equations (56)–(59), the scale of the initial mean velocity and TKE distribution is primarily set by the initial wake diameter $D$. The key time scales of the problem are associated with the mean velocity $((D/U_B)$, the turbulence time scale predicted by the RANS model $(k/\epsilon)$, and the oscillation period due to buoyant forcing (where the buoyancy frequency is given by $N = \sqrt{-(g/\rho_0)(\partial\rho/\partial x_3)}$).

For all of the simulations conducted in this study, the initial wake diameter was given a dimensional value of $D = 1$ m. The gravitational vector was aligned with the $x_3$ axis ($g_3 = -9.81$ m/s). The body propagation velocity $U_B$ was chosen to obtain the desired Reynolds number $Re = U_B D/\nu$. The background salinity stratification $\partial S_B/\partial x_3$ was then set to provide the buoyancy frequency required to match a given internal Froude number $Fr = U_B/ND$.

## 3. Results and Discussion

The employed model variations and the associated equations are summarized in Table 1. RSM1 is a stress transport model employing the realizable, two-component limit pressure–strain model. The variant RSM1a employs an anisotropic model of the dissipation rate tensor $\epsilon_{ij}$. The variant RSM1b employs an anisotropic free-stream turbulence source. RSM2 is a a stress transport model employing the simpler linear pressure–strain model, and EVM1 is the eddy–viscosity model.

The conditions simulated are summarized in Table 2. The scale-resolving simulation studies of Brucker and Sarkar [16] and Dommermuth et al. [14] were used for comparison due to use of the "temporal model", which is more analogous to the $2D + t$ approach than a body-inclusive simulation.

**Table 2.** Summary of the different conditions simulated, with the reference experiment or eddy-resolving simulation.

| Tag | Type | $Re = \frac{U_B D}{\nu}$ | $Fr = \frac{U_B}{ND}$ | $100 \frac{U_{s,0}}{U_B}$ | $100 \frac{\overline{u_i^2}_{CL,0}^{1/2}}{U_B}$ | $100 \frac{\overline{u_i^2}_{\infty}^{1/2}}{U_B}$ | Compare With |
|---|---|---|---|---|---|---|---|
| LP | NZM | 20,000 | 30 | 16 | 14 | 0 | Lin and Pao [24] (LP1979) |
| BS1 | NZM | 50,000 | 4 | 11 | 8 | 0 | Brucker and Sarkar [16] (BS2010) |
| BS1a | NZM | 50,000 | 4 | 11 | 8 | 2 | Brucker and Sarkar [16] (BS2010) |
| BS2 | Drag | 50,000 | 4 | 11 | 8 | 0 | Brucker and Sarkar [16] (BS2010) |
| DOM | Drag | 100,000 | 2 | 11 | 4.5 | 0 | Dommermuth et al. [14] (DOM2002) |

### 3.1. Turbulence Decay

The decay of the root-mean-square vertical-velocity fluctuations (the square-root of the vertical normal stress $\overline{u_3 u_3}$) and root-mean-square scalar fluctuations (the square-root of the scalar variance $\overline{s^2}$) along the wake centerline are depicted in Figures 2 and 3, respectively. The stress transport models achieve the correct decay rate, matching the $(Nt)^{-1}$ exponential decay measured for the experiments detailed by Lin and Pao [24]. The rate is captured for both $\overline{u_3 u_3}$ and $\overline{s^2}$. The differences between RSM1 and RSM2 are trivial for this case. The eddy–viscosity model was not paired with a transport equation for $\overline{s^2}$, and so is omitted from Figure 3.

While reproducing the decay rate, the RSMs systematically under-predict the magnitude of the scalar variance (both the peak value and the value during the decay). A partial explanation of the deficiency may be the uncertainty in the initial conditions for the problem. As noted in the previous section, the scalar-associated turbulence quantities were initially uniformly zero. While computationally convenient, this is clearly non-physical, as mixing of the scalar quantities begins at the onset of turbulence in the body boundary layer, not at a finite downstream distance.

The eddy–viscosity model (EVM1) predicts a much too rapid decay, likely indicating that the expression for the coefficient $c_{\epsilon 3}$ from Henkes et al. [37] is poorly tuned for this particular problem. The rapid extinction of turbulence quantities leads to an unrealistically high preserved wake momentum in the later stages of the wake, as will be discussed in Section 3.2.

Finally, the time-dependent behavior of the centerline value of $\epsilon$ is given for the self-propelled case under the same conditions as the LES of Brucker and Sarkar [16] as Figure 4. The RSMs predict a decay rate in keeping with the LES, while the EVM again predicts a much too rapid decay. The introduction of an anisotropic dissipation rate tensor for RSM1a does not produce a significant difference in behavior, despite the dependence of $\epsilon$ equation coefficients on the stress anisotropy for the RSMs.

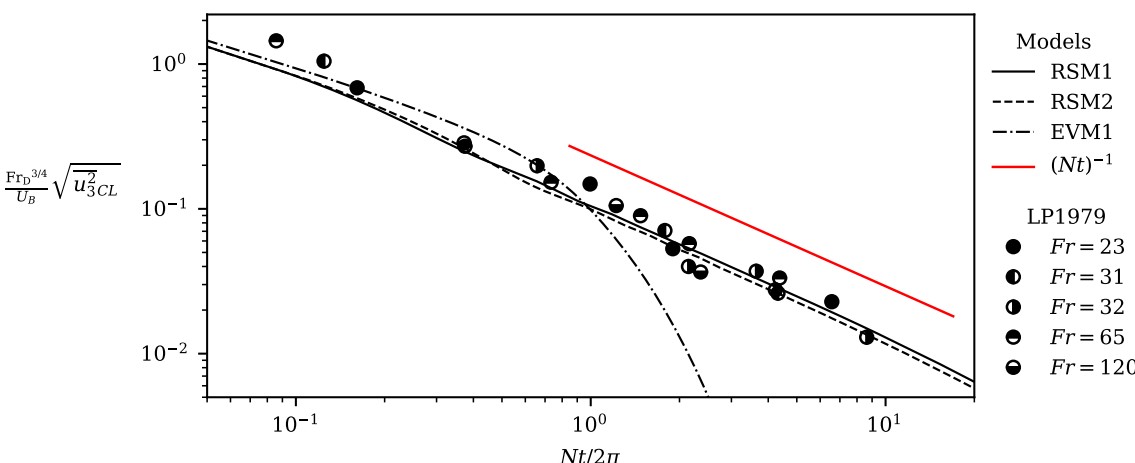

**Figure 2.** Time evolution of the centerline RMS fluctuating vertical velocity for $Re = 20{,}000$, $Fr = 30$ self-propelled stratified wake, with data from Lin and Pao [24] collected at various Froude numbers.

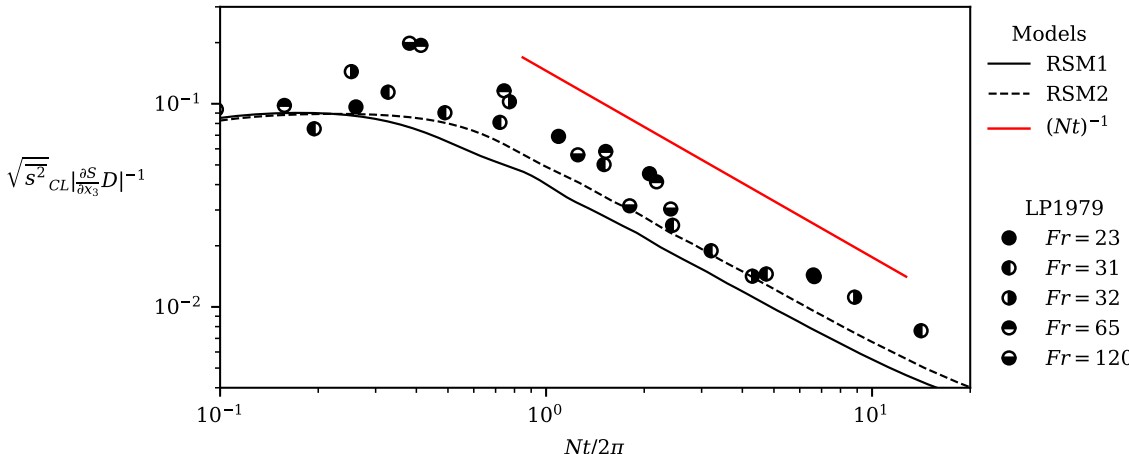

**Figure 3.** Time evolution of the centerline RMS fluctuating scalar for $Re = 20{,}000$, $Fr = 30$ self-propelled stratified wake, with data from Lin and Pao [24] collected at various Froude numbers.

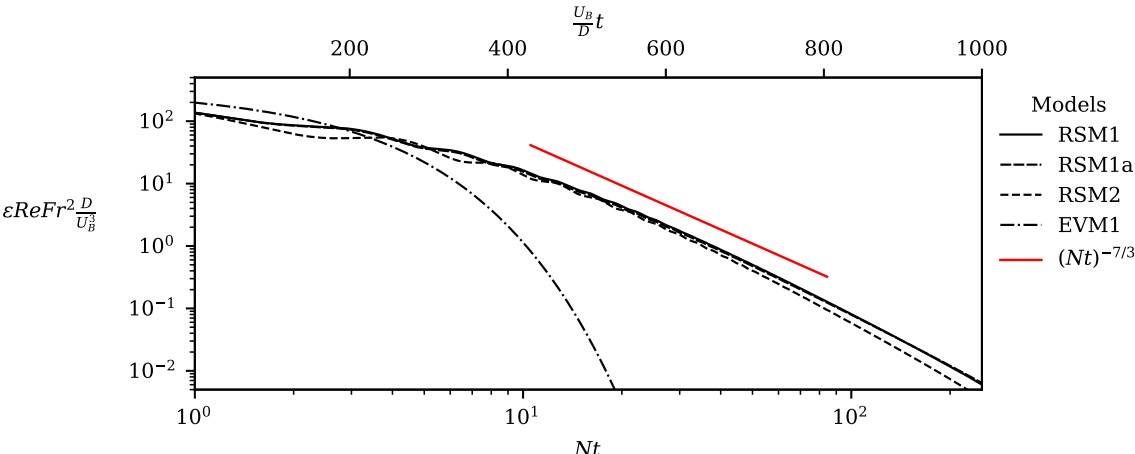

**Figure 4.** Time evolution of the centerline TKE dissipation rate for $Re = 50{,}000$, $Fr = 4$. The $(-7/3)$ exponential decay rate is the same as observed in the LES simulations of Brucker and Sarkar [16].

### 3.2. Wake Momentum Decay

Model predictions of the decay of the mean defect velocity are compared to the scale-resolving simulations of Brucker and Sarkar [16]. Figure 5 shows the comparison for the drag wake. The models all correctly predict the prolonged duration of the momentum wake due to the suppression of turbulent mixing by stratification. As noted in Section 3.1, the rapid extinction of turbulence quantities for EVM1 results in that model predicting a much larger sustained centerline defect velocity.

The RSMs in general reproduce the overall velocity decay well. The RANS models under-predict relative to the scale-resolving model in an approximate region between $Nt \approx 6$ and $Nt \approx 70$. This roughly corresponds to the NEQ region of the wake, according to the stage breakdown suggested by Spedding [18]. Further analysis is required to determine if the disagreement with the scale-resolving simulation can be explained by some physical mechanism occurring in this stage of the wake. As with the model predictions of the turbulence decay, there is not a substantial difference between RSM1 and RSM2 for this metric.

Figure 6 shows the comparison for a self-propelled (NZM) wake. The figure shows both the peak value of the momentum associated with the thrust portion of the wake ($U_s^+$) and the value associated with the drag portion of the wake ($U_s^-$). As with the drag wake, the preservation of the

wake momentum to late $Nt$ is reproduced by the RANS models. EVM1 again predicts a too-high preserved mean momentum.

The RSM predictions for both the thrust and drag portions of the wake are in fair agreement with the scale-resolving model for the NZM condition as well. The under-prediction in the NEQ stage is not present for the self-propelled case. The differences between RSM1 and RSM2 are somewhat more pronounced. RSM1 predicts a slight increase in $U_s^-$ near $Nt = 5$; however, this ultimately puts that model's prediction more in line with the scale-resolved simulation predictions. Finally, it is notable that use of the anisotropic dissipation rate expression in model RSM1a does not produce any qualitative differences in behavior, and appears to reduce agreement with the scale-resolving simulation.

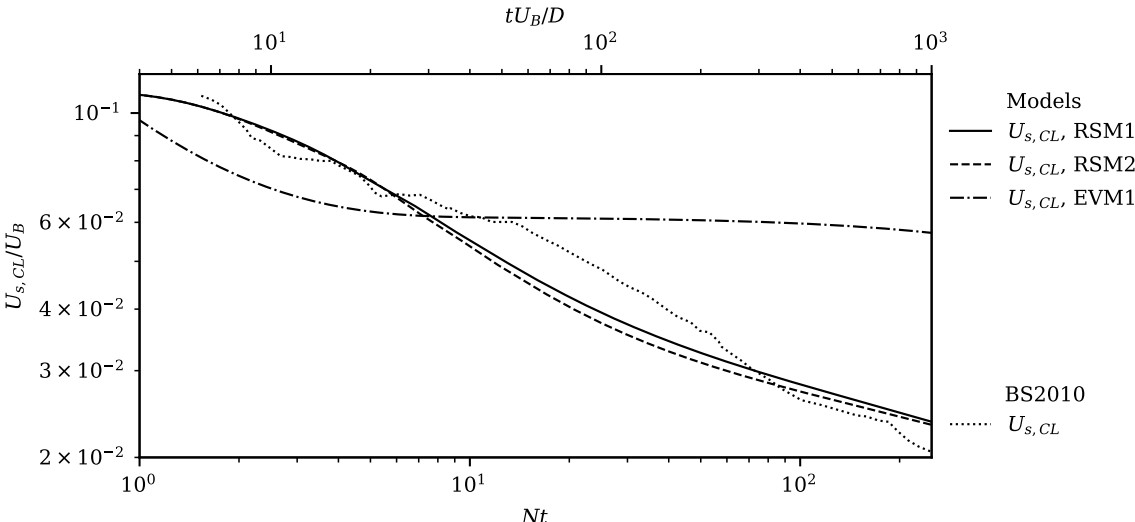

**Figure 5.** Time evolution of the wake velocity defect for the drag wake at $Re = 50,000$, $Fr = 4$. $U_s^+$ indicates the maximum thrust velocity, and $U_s^-$ indicates the maximum drag velocity. With LES predictions from Brucker and Sarkar [16].

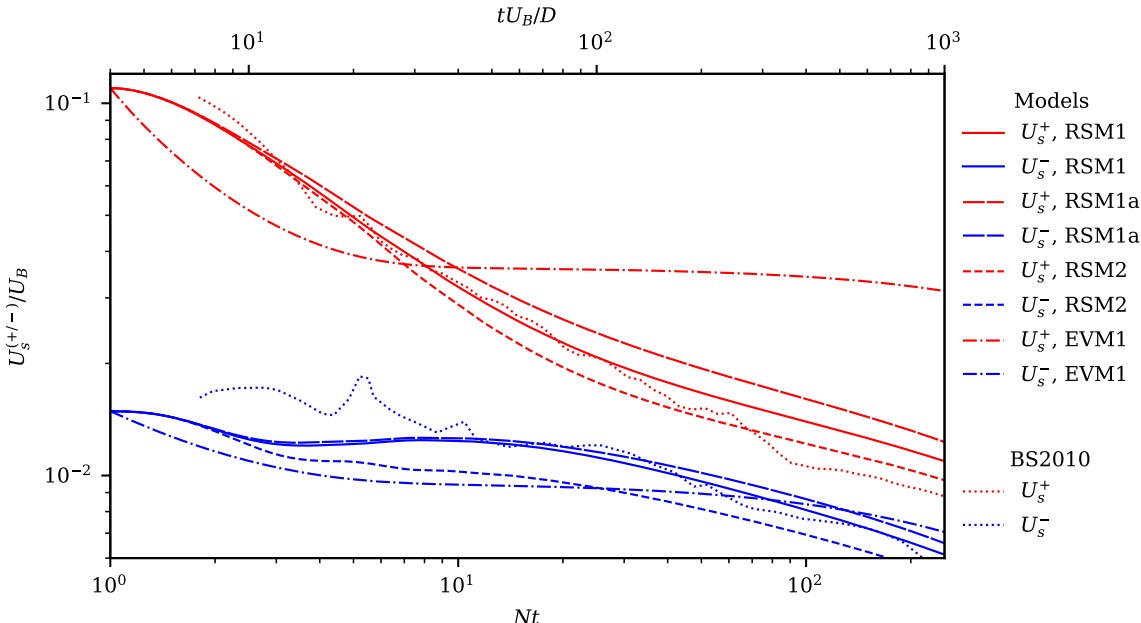

**Figure 6.** Time evolution of the wake velocity defect for the NZM wake at $Re = 50,000$, $Fr = 4$. $U_s^+$ indicates the maximum thrust velocity, and $U_s^-$ indicates the maximum drag velocity. With LES predictions from Brucker and Sarkar [16].

### 3.3. Wake Dimensions

In evaluating model prediction of wake dimensions, the general definition of wake height/width suggested by Brucker and Sarkar [16] is adopted:

$$R_i = 2\frac{\iint U_1^2(x_i - x_i^c)^2 dx_2 dx_3}{\iint U_1^2 dx_2 dx_3}, \qquad x_i^c = \frac{\iint U_1^2 x_i dx_2 dx_3}{\iint U_1^2 dx_2 dx_3} \tag{60}$$

The integrated expression for the momentum width or height allows for direct comparison between the drag and self-propelled cases. Figure 7 shows the model predicted wake dimensions for the pure drag case. The RANS model predictions of the wake growth rate in a horizontal direction roughly agree with the scale-resolving simulation (disregarding the eddy–viscosity model). However, after approximately $Nt = 100$, the RSMs predict a slowing in horizontal growth, which is not observed in the LES results. However, there is substantial disagreement in the predictions of the vertical growth of the wake. Both the RANS and LES approaches predict a local peak in $R_3$ shortly after $Nt = 1$. However, the scale-resolving simulation of Brucker and Sarkar [16] predicts a wake which shrinks in the vertical axis over most of the wake's lifetime, while the RANS models predict a small but positive growth rate. The discrepancy is difficult to explain, and further study is needed to determine the cause of the qualitative difference in behavior.

Figure 8 shows the model predicted wake dimensions for the self-propelled case. The RSM models systematically under-predict the wake width of the self-propelled wake in comparison with the LES; however, the growth rate is in approximate agreement over a portion of the wake lifetime. In contrast with the drag wake, the wake height predictions agree fairly well with the LES, with both the RSMs and the LES indicating a wake which maintains a roughly constant height $R_3/D \approx 0.95$ at late $Nt$. The stronger agreement with LES suggests that the discrepancy seen in Figure 7 may be due to poor initial conditions in the drag wake simulations, rather than a deficiency in the RSMs themselves.

For both the drag and self-propelled cases, the differences between the predictions of RSM1, RSM1a, and RSM2 are again mostly trivial.

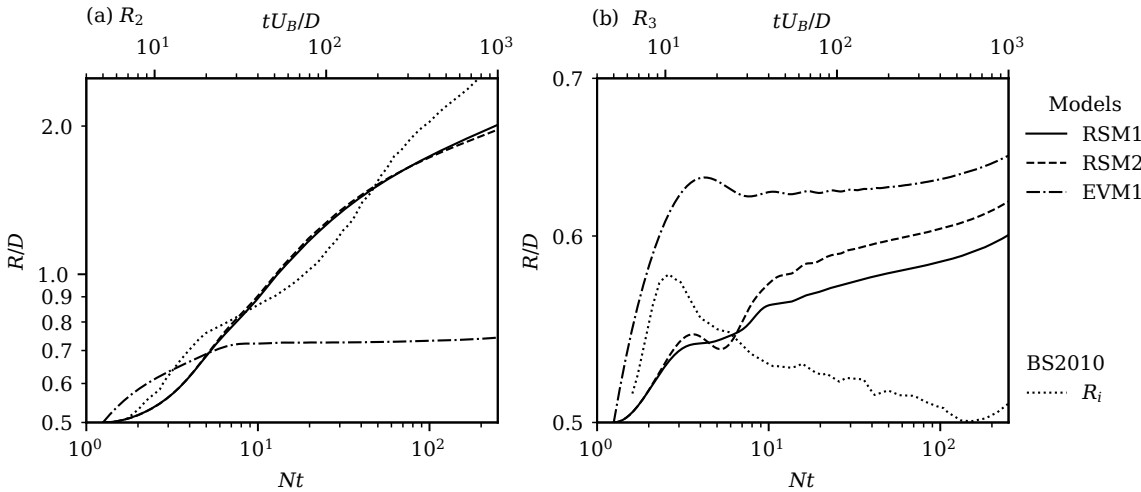

**Figure 7.** Time evolution of the wake width/height based on the integral of the axial momentum. For a self-propelled wake, $Re = 50,000$, $Fr = 4$. With LES predictions from Brucker and Sarkar [16]. (**a**) width ($R_2$), (**b**) height ($R_3$).

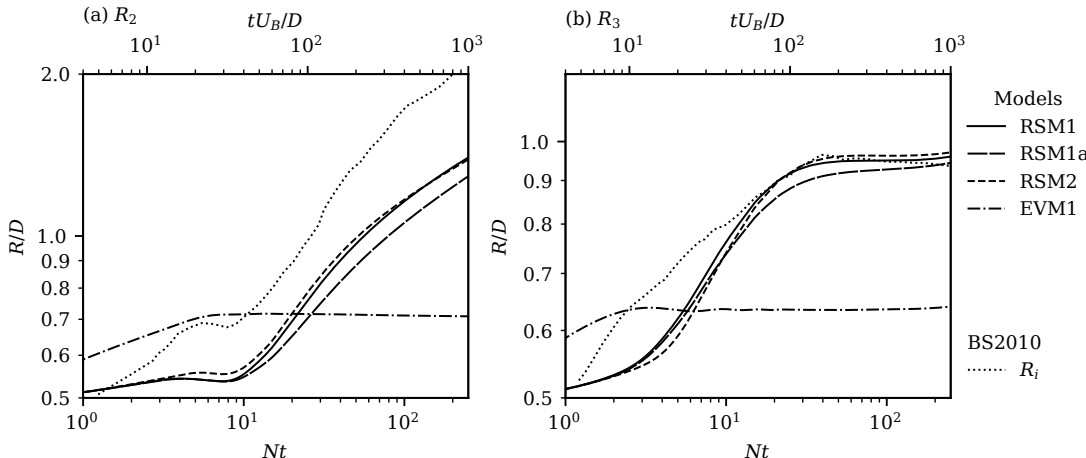

**Figure 8.** Time evolution of the wake width/height based on the integral of the axial momentum. For a self-propelled wake $Re = 50,000$, $Fr = 4$. With LES predictions from Brucker and Sarkar [16]. (**a**) width ($R_2$), (**b**) height ($R_3$).

### 3.4. Spatial Energy Distribution

The spatial distribution of the energy of a wake also be examined; again for comparison with the LES simulations of Brucker and Sarkar [16]. For the drag wake, Figure 9 shows a slice of the domain with the local mean kinetic energy (MKE), while Figure 10 supplies the same for the turbulent kinetic energy. Likewise, Figures 11 and 12 show the MKE and TKE distributions for the self-propelled wake. The predicted MKE and TKE distributions are vertically symmetric.

The drag wake MKE shows a wake which has grown in the horizontal direction, while growth in the vertical is suppressed, in keeping with the thicknesses measured in Section 3.3. Figure 10 shows that at late $Nt$ the TKE has separated into two peaks with a saddle point on the centerline, which is broadly in agreement with the behavior predicted by the LES of Brucker and Sarkar [16]. The primary difference between RSM1 and RSM2 is a slightly larger peak TKE value for the former.

Examining Figure 11, the self-propelled wake possesses two distinct regions of mean kinetic energy. The thrust portion of the wake is still concentrated at the centerline, while the drag portion has been separated into two "lobes" roughly one diameter above and below the centerline. The distribution of MKE compares favorably with the LES of Brucker and Sarkar [16], which predicted similar lobes in similar locations. Figure 12 illustrates perhaps the most significant difference in behavior between RSM1 and RSM2 observed in this study. The former predicts a core of TKE for the self-propelled case, while the latter predicts two separate peaks (similar to the behavior of the drag wake). The prediction of RSM1 is qualitatively more similar to the distribution observed by Brucker and Sarkar [16] for the self-propelled case.

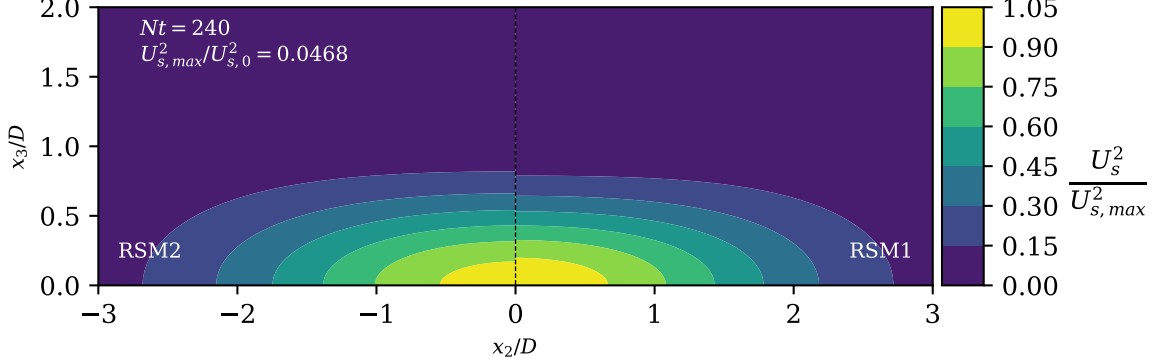

**Figure 9.** Distribution of wake MKE for a drag wake, $Re = 50,000$, $Fr = 4$, $Nt = 240$.

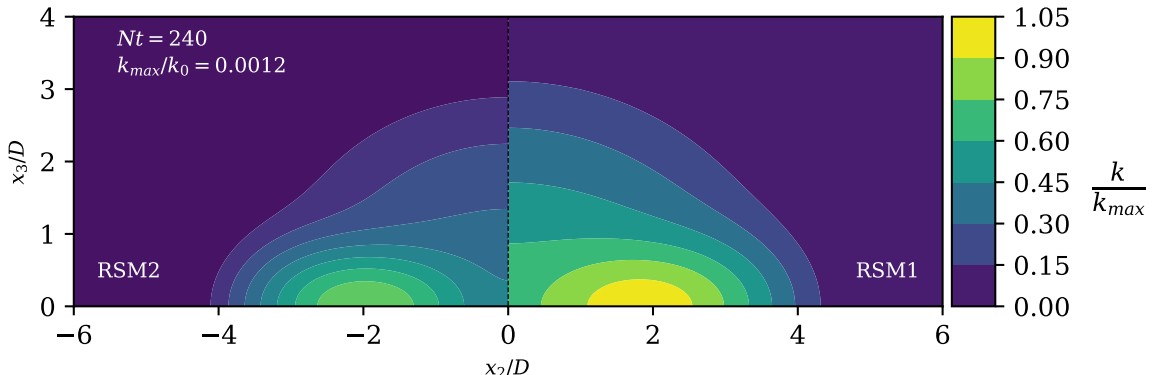

**Figure 10.** Distribution of wake TKE for a drag wake, $Re = 50{,}000$, $Fr = 4$, $Nt = 240$.

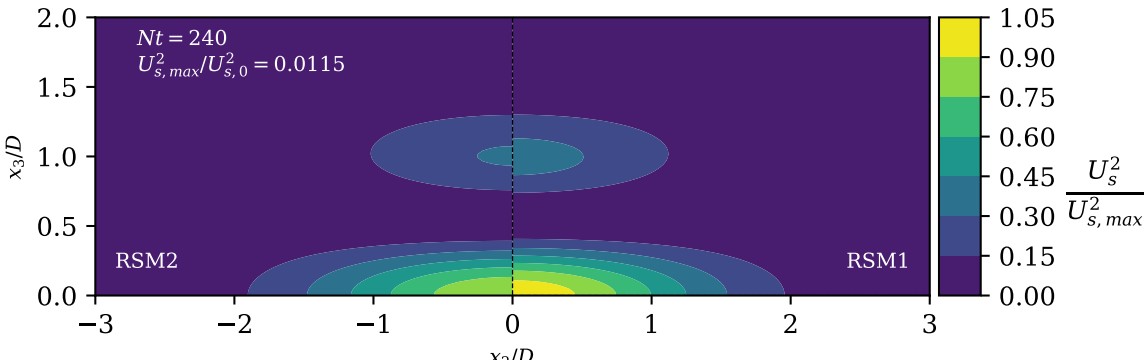

**Figure 11.** Distribution of wake MKE for a self-propelled wake, $Re = 50{,}000$, $Fr = 4$, $Nt = 240$.

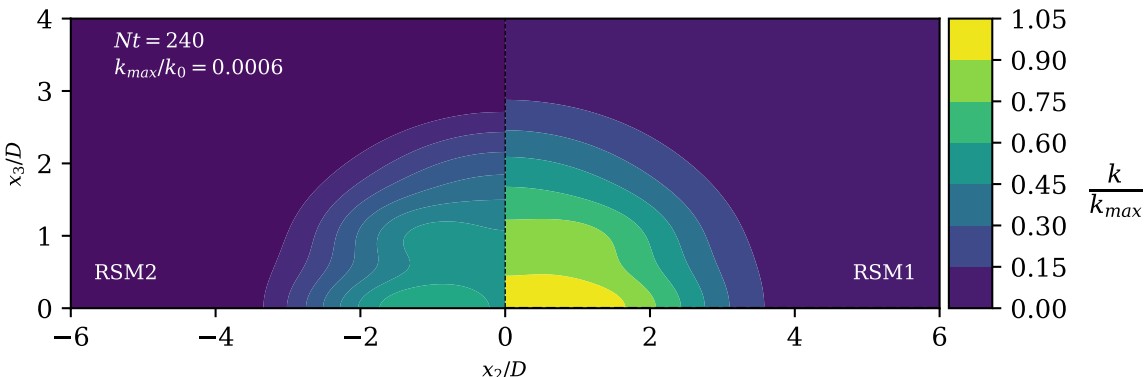

**Figure 12.** Distribution of wake TKE for a self-propelled wake, $Re = 50{,}000$, $Fr = 4$, $Nt = 240$.

*3.5. Collapse-Induced Internal Gravity Waves*

The internal gravity waves (IGWs) produced by the vertical collapse of the wake can be examined taking slices in the $(x_2 - x_3)$ and $(x_1 - x_3)$ planes (note that the $x_1$ direction is taken to be related to $t$ by the body speed for the type of simulation conducted in this study, i.e., $x_1 = U_B t$). Figure 13 shows the perturbation to the salinity field (which, in this case, is equivalent to showing the density perturbation) as a function of time and vertical location. The wake produces waves which propagate upward and downward through the linear stratification, with an oscillation period roughly corresponding to the buoyancy period $2\pi/N$. Figures 14 and 15 show slices in the $x_2 - x_3$ plane, depicting waves which radiate from the wake centerline. The number of rays increases with increasing $Nt$, or equivalently, with downstream distance. It is important to note that both temporal model LES and $2D + t$ RANS models omit body-generated lee waves.

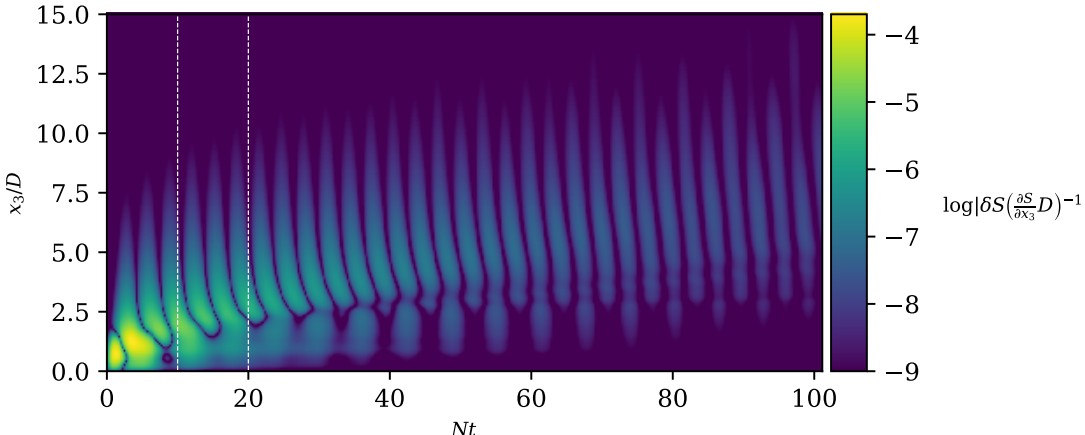

**Figure 13.** Model-predicted density perturbation for a self-propelled wake at $Re = 50,000$, $Fr = 4$, showing the collapse-generated IGWs. The vertical lines at $Nt = 10$ and $Nt = 20$ indicate the locations of the slices shown in Figures 14 and 15.

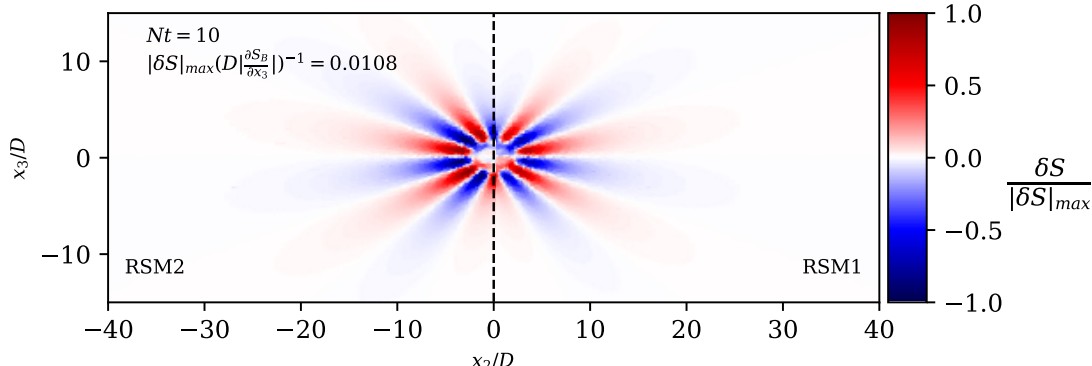

**Figure 14.** Model-predicted density perturbation for a self-propelled wake at $Re = 50,000$, $Fr = 4$, showing the collapse-generated IGWs. $Nt = 10$. There are 14 rays, spaced at roughly $25°$.

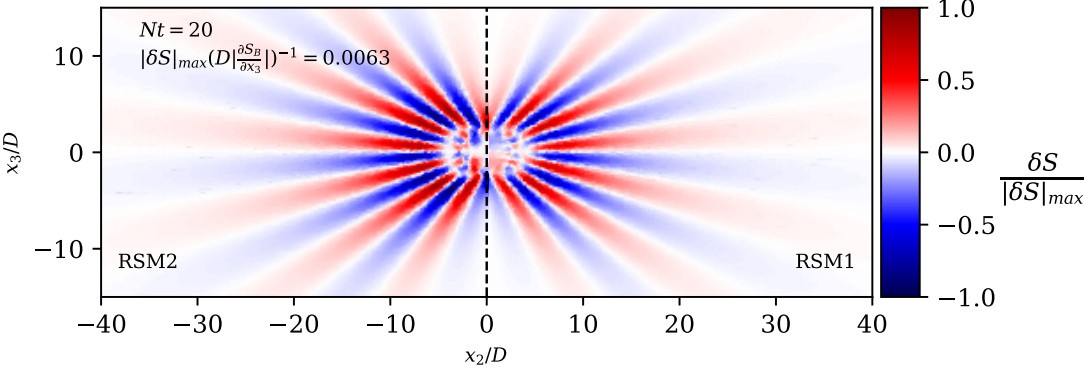

**Figure 15.** Model-predicted density perturbation for a self-propelled wake at $Re = 50,000$, $Fr = 4$, showing the collapse-generated IGWs. $Nt = 20$. There are 26 rays, spaced at roughly $11°$.

*3.6. Integrated Energy Decay*

Figure 16 shows the time variation of the turbulent kinetic and turbulent potential energy (TKE, TPE), integrated over the entire domain. The TKE is separated into vertical (VTKE) and horizontal (HTKE) components. The case is a drag wake under the same conditions as the LES of Dommermuth et al. [14], and the behavior depicted may be qualitatively compared with that study.

The RSM predicts a decay rate roughly in line with the $(-2/3)$ exponential decay measured in the LES over a portion of the wake's lifetime. Additionally, the vertical TKE and TPE oscillate with a period roughly equally to the buoyancy period $2\pi/N$, exchanging energy as they do so. The oscillatory behavior is also in line with the behavior observed by Dommermuth et al. [14].

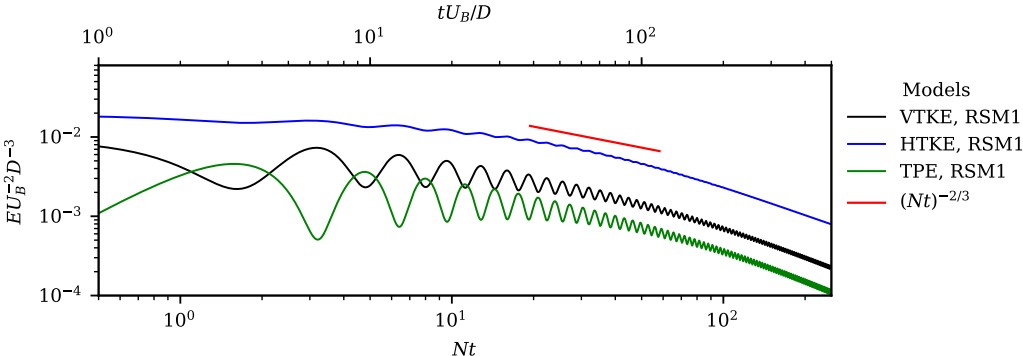

**Figure 16.** Model-predicted time evolution of integrated wake vertical and horizontal turbulent kinetic (VTKE, HTKE), and TPE for a drag wake at $Re = 10^5$, $Fr = 2$. The $(-2/3)$ decay is that predicted by the LES simulations of Dommermuth et al. [14].

### 3.7. Free-Stream Turbulence Effects

Finally, the effect of increased free-stream turbulence intensity may be briefly explored by increasing the strength of the sustaining sources included in the turbulence models. As indicated in Table 2, a second set of simulations was conducted with a high free-stream TI for the self-propelled case (case BS1a). The test was conducted for both an isotropic free-stream $\overline{u_i u_j}$ source, and the anisotropic source given by (54). Figure 17 shows the decay of the mean velocity. Strong background turbulence predictably increases the rate at which the mean velocity decays. The anisotropic source term appears to produce a stronger effect for the same free-stream TI.

Figure 18 depicts the predicted wake dimensions under the same conditions. In this case, the sustaining sources are strong enough to overcome the buoyancy effects, and the wake grows in both the vertical and horizontal direction. The growth continues past the point at which the wake height ceases growth in quiescent conditions. By late $Nt$, the wake turbulence has been reduced to the background value, and turbulent transport of the wake momentum is exclusively by the background turbulence.

While the predicted behavior shown in Figures 17 and 18 makes intuitive sense, further study is required to confirm the efficacy of the free-stream source approach employed.

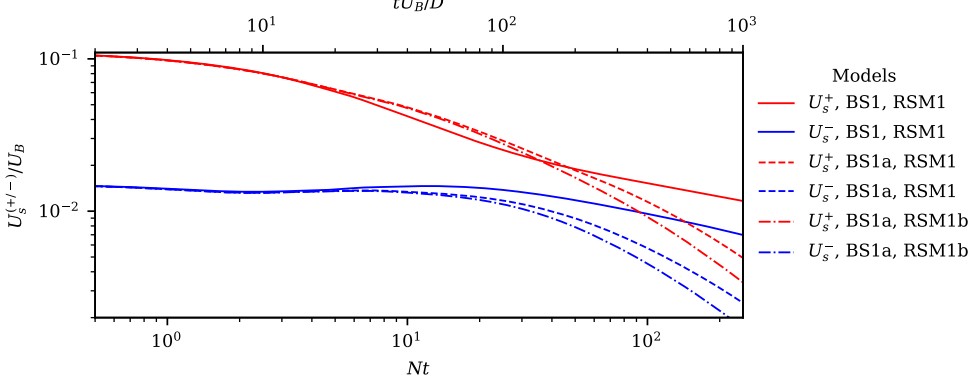

**Figure 17.** Time evolution of the wake velocity defect for the self-propelled wake at $Re = 50,000$, $Fr = 4$, with a free-stream turbulence intensity of 2%. $U_s^+$ indicates the maximum thrust velocity, and $U_s^-$ indicates the maximum drag velocity.

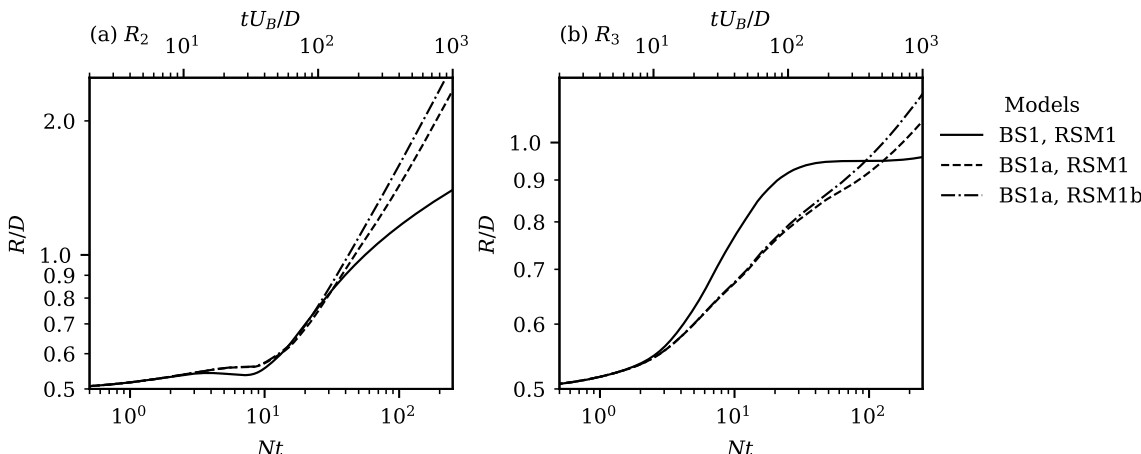

**Figure 18.** Time evolution of the wake dimensions for a self-propelled wake at *Re* = 50,000, *Fr* = 4, with a free-stream turbulence intensity of 2%.

## 4. Conclusions

In this work, we have demonstrated the use of a pair of anisotropic stress-transport RANS turbulence models intended to be used to simulate full-scale wakes in an active ocean environment. The models were found to reproduce a number of important stratified wake behaviors as observed in LES and laboratory studies. In particular, the models capture the decay rates of key turbulence quantities, the preservation of mean momentum to late *Nt* due to suppression of turbulence, and the wake collapse and internal gravity wave production. It was found that the more complex TCL pressure–strain based RSM did not differ significantly in behavior from the simpler linear model under the conditions simulated. Likewise, the use of an anisotropic dissipation-rate tensor for the stress transport equations did not substantially improve agreement with LES model predictions. Use of these more complex models required approximately 10–20% more computing times over the linear RSM for a given wake case. Further study at late *Nt* is needed to determine if the TCL model's additional cost is justified for low turbulence Froude numbers. Application of the models to late *Nt* will also likely require other modifications; the wake approaches both low turbulence Reynolds number and low Froude number conditions, likely beyond the range of validity for the models as implemented here (which were developed for high Reynolds number turbulence). Finally, the models were further modified with additional source terms to supply a nonzero background turbulence, which was found to increase the rate of wake decay and increase wake thickness growth. More tests are needed to carefully validate this capability.

**Author Contributions:** Methodology, software, writing—original draft preparation, D.W.; software, supervision, project administration, writing—review, and editing, E.P. All authors have read and agreed to the published version of the manuscript.

**Funding:** This research was funded via contract with a corporate partner.

**Acknowledgments:** Virginia Tech Advanced Research Computing and the Department of Defense High Performance Computing Modernization Program are acknowledged for providing supercomputing resources. The authors are also thankful to K.J. Moore, President of Cortana Corporation, for his continuing support.

**Conflicts of Interest:** The authors declare no conflict of interest.

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
