# Peer review of "Anisotropic RANS Turbulence Modeling for Wakes in an Active Ocean Environment†"

_fluids, doi:10.3390/fluids5040248_

Round 1

Reviewer 1 Report

This work investigates the performance of anisotropic stress-transport RANS in capturing the dynamics of wakes in the presence of stratification in oceanic environment. The new RANS results are compared with those from LES and laboratory experiments, and good agreements are reported. The paper is well-written, and the idea of improving cost-effective models (such as RANS) to correctly capture the flow dynamics and predictions as are in LES and measurements, is very interesting. However, there are some points and statements in the manuscript that are not clear (and are even confusing) that make the outcome of this work to be a little overweighted in comparison with more advanced studies such as LES —while I agree that LES and DNS runs are computationally expensive— and therefore, I would like to ask authors to address these issues before I can recommend the publication of this manuscript in Fluids. With these modifications, I believe this paper will be well placed as a cost-effective model in communities for ocean sciences and numerical simulations. Here, I have itemized my main concerns followed by minor comments: 

  1. Equation (40): how do you choose these correlation coefficients? In the manuscript, there is not much information that how these coefficients are consistent with the dynamics / physics of the flow. This brings the idea that the coefficients are set arbitrary, which can cause biases to the RANS modeling. I would like to ask authors to add some explanations to the manuscript about their procedures to select transport coefficients. Similar comments hold for equation (47). 
  2. Line 244-246: Brucker and Sarkar [16] employ LES approach in their studies. LES uses a scale-aware parameterizations for model closure. I think the argument that fluctuations do not change the dynamics of wake substantially, is tested in LES runs, but not in RANS, which is not a scale-aware model, and only resolve the mean state. I believe the authors should justify this assumption in RANS before using it.  

Minor comments:

  1. Line 22 (also elsewhere in the paper and abstract): the word "eddy-resolving" is mainly developed to be used for large-scale atmosphere and ocean models (AOM) for resolving mesoscale eddies. I suggest the authors use another terminology here for wake studies. For example, they can use the word "scale-aware" or "resolved-scale" instead. 
  2. Line 65: In the literature of LES, the anisotropic features of flow motions due to the presence of stratification can be captured by resolved scales if the grid spacing revolves the buoyancy scale L_b (as in Khani & Waite 2014, J Fluid Mech 754, PP 75-97; Khani & Waite 2015, J Fluid Mech 773, pp 327-344; Khani & Waite 2020, Mon Wea Rev 148, pp 4299-4311). I recommend authors to include these work in their literature review section to include more studies that consider the effects of stratification in active atmosphere and ocean environment. 

Author Response

Responses to the reviewer's primary concerns are given below: 1 - The coefficient values and parameterizations for the scale equation (40) are those employed by Craft, Launder, and others in a variety of works, and are used without modification or additional problem-specific tuning. The form of the equation was tested by those authors for a variety of free-shear flows, including jets and wakes (though not subjected to a body force, as in this work); as we are here applying the model to a free-shear flow, a re-tuning of the model was not undertaken. Similarly, the values for equation (47) are those of the model developed by Henkes, et al. The manuscript will be edited to make this more clear. Additionally, it is acknowledged that the form of the scale equation is a potential area for significant model improvement, which is a possible avenue for future work. 2 - As noted in the manuscript, the turbulence quantities associated with scalar fluctuations were originally initialized both as zero-valued and using algebraic expressions dependent on the turbulence kinetic energy, scale variable (epsilon), and background gradient. The algebraic expressions were reduced forms of the transport equations for the turbulence quantities, under the assumption of equilibrium. However, it was ultimately decided not to employ the algebraic expressions for the presented simulations, primarily because the simulations in this work are compared with LES in which the scalar turbulence quantities were initialized to zero. Simulating the same problem but with different initial conditions would not have been an effective test of the RANs model capabilities. Additionally, the difference between simulations run with algebraic expressions and those with zero initial conditions were found to occur only in a brief initial period less than Nt=1. Quantities of interest such as the decay rates of turbulence and energy quantities, wake dimensions, and mean wake velocity were found to be minimally impacted in the stages of the wake for which comparison with LES or experiments was conducted. The manuscript will be modified to more clearly communicate this decision making process. Minor comments: 1 - The vocabulary employed in the manuscript will be modified as suggested to avoid confusion. 2 - The additional reference recommendations are greatly appreciated! We will add them to the review section of the manuscript.

Reviewer 2 Report

Anisotropic RANS turbulence models are tested for wakes in an active ocean environment. A full discussion of the closure equations is carried out in a very instructive form. Simulations are performed with OPENFOAM.In particular, the models capture the decay rates of key turbulence quantities, the preservation of mean momentum to late Nt due to suppression of turbulence, and the wake collapse and internal gravity wave production. The models were further modified with additional source terms to supply a nonzero background turbulence, which was found to increase the rate of wake decay and increase wake thickness growth. The authors claim, and I agree with them, that more tests are needed to get full verification.

The article is very well written and I have no major issues. I am happy to advise it for publication in Fluids. I have, however, some minor suggestions:

  • Add a simple sketch of the problem where the main scales are depicted.
  • Help the reader to understand the context and main physics features of the problem.
  • Highlight what is really new in the paper.
  • Add a presentation of the paper structure at the end of the introduction.
  • Line 151 and 153: what some models refer to?
  • Have closure models a well defined limits of validity? Please specify.

Author Response

Responses to the reviewer's comments are given below:

1 - The diagram of the initial conditions (Figure 1) for the RANs model simulations could serve in the capacity which the comment suggests, given the proper context. The text of the manuscript will be edited with additional commentary on how this figure relates to the scales of the problem.

2 - The language at the beginning and end of the paper will be modified to more clearly indicate the novel contributions (primarily, the application of the realizable RSM to stratified wakes and the inclusion of environmental sustaining sources). The introductory section will also be edited to include a presentation of the paper structure as suggested.

3 - The components of the velocity gradient tensor (equations (25) and (26)) are employed primarily by the realizable pressure strain model (equation (34)) and the eddy-viscosity model (48). The text of the manuscript will be modified to make this more clear.

4 - The forms of the RANs models employed are primarily applicable to high Reynolds number flows, and would required modification for lower Re flows.  One of the goals of this study was to determine if the RANs models employed can be effectively applied to wakes in the low Froude number regime previously examined by others using LES and experimental techniques. A discussion of the range of validity in the Re-Fr parameter space will be added in the concluding remarks of the manuscript.

Round 2

Reviewer 1 Report

My major concerns were mostly addressed in the revised manuscript by adding extra clarification about the model coefficients. I am also ok with the revisions on my minor comments. I can now recommend the publication of this manuscript in Fluids. 

Reviewer 2 Report

I am happy to advise the publication of this paper as ti is in Fluids